# Lahar events in the last 2,000 years from Vesuvius eruptions. Part 3: Hazard assessment over the Campanian Plain

Laura Sandri[1], Mattia de' Michieli Vitturi[2], Antonio Costa[1], Mauro A. Di Vito[3], Ilaria Rucco[4], Domenico M. Doronzo[3], Marina Bisson[2], Roberto Gianardi[2], Sandro de Vita[3], Roberto Sulpizio[1,5,6]

[1]Istituto Nazionale di Geofisica e Vulcanologia, Sezione di Bologna, Bologna, Italy
[2]Istituto Nazionale di Geofisica e Vulcanologia, Sezione di Pisa, Pisa, Italy
[3]Istituto Nazionale di Geofisica e Vulcanologia, Osservatorio Vesuviano, Napoli, Italy
[4]School of Engineering and Physical Sciences, Heriot-Watt University, Edinburgh, UK
[5]Dipartimento di Scienze della Terra e Geoambientali, University of Bari, Bari, Italy
[6] IGAG CNR, via Mario Bianco 9, Milano, Italy

*Correspondence to*: Laura Sandri (laura.sandri@ingv.it)

**Abstract.** In this study we present a novel general methodology for Probabilistic Volcanic Hazard Assessment (PVHA) for lahars. We apply the methodology to perform a probabilistic assessment in the Campanian Plain (Southern Italy), focussing on syn-eruptive lahars from a reference-size eruption from Somma-Vesuvius. We take advantage of new field data relative to volcaniclastic flow deposits in the target region (in a companion paper, Part 1) and recent improvements in modelling lahars (in the other companion paper, Part 2). The former allowed defining proper probability density funtions for the parameters related to the flow initial conditions, and the latter allowed computationally faster model runs. In this way, we are able to explore the effects of uncertainty on the flow initial conditions on the invasion of lahar in the target area, by sampling coherent sets of values for the input model parameters and performing a large number of simulations.  We also account for the uncertainty on the position of lahar generation, by running the analysis on 11 different catchments threatening the Campanian Plain. The post-processing of the simulation outputs led to the production of hazard curves  for the maximum flow thickness reached on a grid of points covering the Campanian Plain. By cutting the hazard curves at selected threshold values, we produce a portfolio of hazard maps and probability maps for the maximum flow thickness. We also produce hazard surface and probability maps for  the simultaneous overcoming of pairs of thresholds in flow thickness and dynamic pressure. The latter hazard products represent, on one hand, a novel product in PVHA for lahars, and, on the other hand, a useful means for impact assessment, assigning a probability to the occurrence of lahars that have simultaneously a relevant flow thickness and large dynamic pressure.

## 1 Introduction

Lahars are flows of water and entrained sediments that originate from the remobilization of volcaniclastic deposits by water, either from rain, melting ice or snow, or a sudden release by a crater lake (see the companion paper by de' Michieli Vitturi et al., this issue, and references therein). They represent one of the highest death-toll processes among volcanic phenomena. According to the analysis by Auker et al. (2013), syn-eruptive lahars are responsible for about 14% of the fatalities in their

database, whereas post-eruptive or inter-eruptive lahars cause another 3%. Among the most tragic episodes of lahar impact, we recall the lahar generated from Nevado del Ruiz, which buried the city of Armero killing about 23000 people, making it

the second-worst volcanic disaster of the twentieth century (Pierson et al, 1990, Voight et al, 2013, Parra and Cepeda, 1990). Other examples include the series of lahars that hit the surroundings of Pinatubo volcano in the years after the 1991 eruption (Pierson et al, 1996, Umbal and Rodolfo, 1996, Rodolfo et al, 1996),  the Tangiwai disaster at Ruapehu (Manville, 2004) and the lahars from the 2008-2009 Chaiten eruption (Pierson et al, 2013).

A simplified method commonly used so far to describe lahar impact has been LAHARZ model (Schilling, 1998). It is based

on the statistical correlation between the inundated area and the mass flow volume inferred from past events. However, in recent years, a few examples of probabilistic hazard assessment for lahars based on more robust statistical treatments, like statistical surrogates or emulation approaches, have been proposed for different volcanoes worldwide, such as Mead and Magill (2017) on Ruapehu (New Zealand), Tierz et al (2017) on Vesuvius (Italy), Gattuso et al (2021) on Vulcano (Italy).

Hazard assessment for lahars needs to consider: i) the identification of potential source regions for volcanic material and for

water, including snowcaps and glaciers; ii) the potential magnitude and characteristics of the flow; iii) the topography between the source region and the potential targets at risk; iv) the potential for modification of the flow properties along the path; v) the frequency of such events in the past; vi) the meteorological data at the source region and along the potential path of such flows, especially for extreme events.

As explained in the companion paper by de' Michieli Vitturi et al. (this issue), lahars can change character downstream,

through processes of flow bulking and debulking, generating high variability in both time (i.e. unsteadiness) and space (i.e. non-uniformity) for variables pivotal for hazard assessment, such as particle concentration, granulometry and componentry, bulk rheology, and velocity. The full complexities associated with these processes prevent our ability to effectively model these flows for quantitative hazard assessment purposes. At present, even if we could describe all the underlying physics, a full 3D simulation of all these phenomena would require prohibitive computational costs, and current numerical models

describe only a part of the observed phenomena or use simplified approaches (see the companion paper by de' Michieli Vitturi et al., this issue). In terms of numerical modelling, a good compromise between the completeness of the physics behind these phenomena and the computational feasibility is represented by the shallow water approach (de' Michieli Vitturi et al., this issue), where model complexity is reduced with a depth-averaging of flow properties. This approach approximates the original 3D problem with a 2D model, and it is the one we apply in this work.

In the Campania region, which is largely covered by fallout and pyroclastic density current (PDCs) deposits from eruptions of Somma Vesuvius, Campi Flegrei, and Ischia volcanoes, the signature of several syn- and post-eruptive lahars has been found in the geological record (Di Vito et al, this issue, Sulpizio et al., 2006; Zanchetta et al., 2004a,b). Furthermore, detailed lists of documented lahars in the 20th century are available in the literature (Fiorillo & Wilson, 2004). Despite such evidence, up to now most of the Probabilistic Volcanic Hazard Assessments (PVHA) for this region has mostly focussed on

PDCs (e.g., Neri et al., 2008; Gurioli et al, 2010; Neri et al., 2015; Sandri et al., 2018; Tierz et al, 2017) and tephra fallout (e.g., Costa et al., 2009; Selva et al., 2010; Sandri et al., 2016; Selva et al., 2018; Massaro et al, 2023), while systematic quantitative hazard assessments from lahars (see for example Jenkins et al., 2022) have been lacking. An exception is provided by Tierz et al. (2017), who applied a Bayesian Belief Network to assess the effect of different factors (linked to rainfall intensity and volcanoclastic volume) on the probability of different initial volumes of lahars. However, that study did

not explore the variability in the hazard assessment related to the initial flow conditions (mostly linked to the flow volume, detachment area, and volumetric solid fraction).

PVHA for lahars needs performing a high number of simulations in order to enable a quantification of the uncertainty linked to model parameters.

Recent technical improvements (e.g. code parallelization) and generalisations (e.g. description of erosion and deposition

during the flow) of lahar models, such as that implemented in the most recent version of the IMEX-SfloW2D model, described in the companion paper by de' Michieli Vitturi et al. (this issue), permit the simulations of hundreds of simulations from different catchments, with different initial and boundary conditions in reasonable times, necessary for characterising the intrinsic variability and for the production of hazard maps.

Following several surveying campaigns carried out for characterising lahar deposits in natural exposures, archaeological

excavations, and ad hoc trenches in the plain surrounding the Vesuvius edifice and along the Apennine valleys, Di Vito et al (this issue) present the results of a multidisciplinary study which shows the presence of volcaniclastic deposits (mostly debris and mud flows but also from hyperconcentrated flood flows) even in areas very far from both the Apennine hills and the valleys of Somma-Vesuvius, demonstrating the high mobility of these flows. In particular Di Vito et al (this issue) focused on the analysis of the syn- and post-eruptive lahar deposits generated by the two sub-Plinian eruptions of Vesuvius, 472 AD

and 1631. Thicknesses, sedimentological features (lithic content, pumice provenance, grain size, boulder entrainment) and vertical/lateral continuity of the deposits were reported during the campaigns, in order to establish characteristic facies (massive-to-structured, poorly- to better-sorted with respect to the primary pyroclastic deposits and topography) and general flow dynamics (velocity, dynamic pressure, thickness) of those volcaniclastic systems. It results that the inclusion of the fine ash into the whole deposits distribution, the depositional mechanism of the primary pyroclastic deposits (fallout vs. current),

and the large-scale topographic effects (plain vs. valley) are the main geological features affecting the size and style of the remobilization occurred for the two eruptions (Pollena and 1631).

In this work, we take advantage of these new field data analyses and recent improvements in modelling lahar flows (de' Michieli Vitturi et al., this issue), to explore the effect of uncertainty on the flow initial conditions on the invasion of lahar in

the Campanian Plain (Fig. 1a), by sampling coherent sets of values for the input model parameters, and subsequently performing a considerable number of lahar simulations (1100 in total) needed for the production of hazard maps.

We present a novel general methodology for PVHA for lahar flows but we focus on syn-eruptive lahars, conditional to the occurrence of a reference eruption for Somma-Vesuvius (that is the Medium Magnitude Scenario by Cioni et al., 2008; Macedonio et al., 2008; Sandri et al., 2016).

In particular, we account for:

- 11 different catchments (Fig. 1b) where lahars could originate and impact the target area of the Campanian Plain, both from the Somma-Vesuvius edifice and from the Apennines sectors to the East and South;
- deposits from PDCs (mostly on the Somma-Vesuvius catchments) and from tephra fallout (on the Apennines catchments) from the reference eruption of Somma-Vesuvius;

•  the maximum expected rainfall in a few days, taken on the order of 500 mm, as extracted from the rainfall record in the last 70 years in the Campania Region (Fiorillo & Wilson, 2004);coherence among the initial values of the flow (initial thickness, detachment area, and volumetric solid fraction), the deposit porosity (water-saturated) and the available water from rainfall and volcaniclastic sediments from the reference eruption. In order to do so, we build up a strategy to sample the input model parameters, which will be illustrated in Section 3.1.

The first and latter points, in particular, allow us to explore the uncertainty on the position of lahar generation, and on the flow initial conditions (in terms of area of detachment, initial volume, volumetric solid fraction).

The goal of the study is to quantify the conditional probability of invasion by at least one lahar originating from the remobilization of tephra deposits due to the reference eruption at Somma-Vesuvius (Sandri et al., 2016). In order to provide useful results for future quantification of the impact of syn-eruptive lahars, we express our results in terms of exceedance

probability for selected thresholds of two variables pivotal for hazard assessment, such as maximum flow thickness and dynamic pressure. The domain of interest, covering the Campanian Plain (see Fig. 1a), was discretized, for computational reasons, on a 50 m × 50 m grid. This resolution represents a good compromise between solution accuracy and computational time required for a simulation (see companion paper, de' Michieli Vitturi et al., this issue). As we mentioned above, we also compute the exceedance probability of selected pairs of critical thresholds in dynamic pressure and flow thickness, that

jointly are key parameters for lahar impact assessment (e.g., Wilson et al, 2014). In other words, we take into account the flow "history" in every target grid point, and we compute the probability of the flow to simultaneously exceed (in at least one time step of the simulated flow) two values of flow thickness and dynamic pressure, repeating this for several pairs of values.

The paper is organised as follows: first we very briefly summarise the geological information and the features of the model,

second, we present the method used for the PVHA, and then, we show the results as maps. Finally, we present a Discussion and conclusion in order to highlight the main achievements and the current limitations, which will be addressed in future works.

## 2 Field data and transport model

The code developed by generalising the ImexSFlow-2D model for describing lahar flows, described in the companion paper

by de' Michieli Vitturi et al. (this issue), was calibrated using the field data presented by De Vito et al. (this issue). The field data were used also to define the available initial deposit from a reference eruption from Somma-Vesuvius in the different catchments, from PDC and tephra fallout deposits.

### 2.1 Remobilizable PDCs and tephra deposits

In this study, for hazard assessment purposes we considered a reference eruption belonging to the Medium Magnitude

Scenario (MMS; Sandri et al., 2016). The initial volumes that can be remobilized as lahars come from PDC and tephra fallout deposits and from the available water (rain in our case). PDC deposits are more relevant for Somma-Vesuvius

catchments, which are proximal to the source; tephra fall deposits, dispersed by the local wind fields, are mostly relevant for the Apennines catchments, where PDCs from a MMS eruption do not leave any appreciable deposit.

As regards the PDC deposits, we used the field data from the most recent subplinian eruptions, which are the 472 CE
(Pollena; Sulpizio et al, 2005) and the 1631 CE (Rosi et al, 1993) eruptions. Cautiously in each grid cell of a given catchment we considered as available PDC deposit the maximum thickness between those from these two PDC events, as mapped by Gurioli et al. (2010).

As regards tephra fall deposits, we rely on the results of the simulations presented in Sandri et al. (2016), where 1000 simulations were performed for the MMS considering variability of the meteorological conditions and ESPs (total erupted
mass, mass eruption rate, total grain size distribution). Specifically, we randomly sampled (without repetition) 100 fallout deposits from the 1000 available. Hence we used those deposit distributions for the simulations performed with the generalised IMEX-SfloW2D model.

## 2.2 Lahar runout

A considerable set of lahar runout estimations inferred from the deposits in the Campanian Plain (see Fig. 2), associated to
the 472 and 1631 eruptions (Di Vito et al., this issue), were used to calibrate the empirical parameters needed for friction, erosion, and deposition terms (de' Michieli Vitturi et al., this issue).

## 2.3 Grain size distribution

The Grain Size Distribution (GSD) is another critical input for modelling lahar transport (de' Michieli Vitturi et al., this issue). For this study we used those reconstructed by Di Vito et al. (this issue) based on the field data from the catchments
around Somma-Vesuvius (catchments 1 to 6 in Fig. 1b) integrated with those from Pozzelle quarry (Sulpizio et al., 2006), and from Vallo di Lauro valley (at the basis of catchment 8 in Fig. 1b). In particular, both types of GSDs have been reconstructed by fitting a mixture of two Weibull distributions and then to average the values of the local grain sizes sampled in the field at the basis of these catchments, separately. The reconstructed GSDs are reported in Fig. 3: for the Apennine catchments (7-11 in Fig. 1b) we used the GSD reconstructed from the field data taken in Vallo di Lauro (Fig.3b), while for
the Vesuvian catchments (1-6 in Fig1b) we obviously use the GSD reported in Fig.3a and reconstructed on field data taken there.

Finally, in order to assess the effect of the uncertainty on the reconstructed GSDs we performed a sensitivity test (see Supplementary Material 1) on the weight of the coarse and fine populations on the simulated deposits, showing that is not very critical in terms of simulated deposit, maximum thickness and dynamic pressure after 24 hours from the flow
mobilization.

## 2.4 Digital Elevation Model

For a correct modelling of the areas invaded by lahars it is necessary to use a Digital Elevation Model (DEM) as accurate as possible. To this end we used a Digital Terrain Model (DTM) derived from an airborne LiDAR survey of 2012 combined with the TIN Italy Topography (Tarquini et al., 2007) in a portion of the sub-Apennine areas where the LiDAR data were not available. The LiDAR data were provided by the Italian Ministero dell'Ambiente e della Tutela del Territorio e del Mare (MATTM) through a series of ASCII files storing the elevation data in the latitude-longitude WGS84 reference system. The tiles, each covering 1 km2, have been processed to create a single elevation matrix geo-referenced in the WGS84-UTM-Zone 33N geodetic cartographic system and memorising the elevation data at 32 bit. The obtained matrix, with spatial resolution of 1 m and vertical accuracy < 30 cm (Pizzimenti et., al 2016), was cleaned from residual anthropic or artificial features and subsequently resampled at 10 m cell size in order to be combined with the TIN Italy model. The resulting matrix (4129 × 5088 cells), covering 1600 km2, was used as the topography model for the area (Fig. 1c) for the simulations, discretized at a computational grid of 50 m × 50 m, which was tested by de' Michieli Vitturi et al. (this issue) as a good resolution able to reproduce the main features of the flow in reasonable computational times.

## 3 Methods

### 3.1 Sampling strategy

In order to explore the natural variability in the processes governing lahar initiation, we first identify the key independent parameters. Given a catchment, the key parameters are the initial flow volume and initial solid volumetric fraction ($\alpha_s$); the first depends on the initial area of lahar detachment, on the thickness of the available remobilizable deposits, and on the available water from both rain or other external water sources and water content filling soil pores.

The initial value of $\alpha_s$ is very variable and hard to assess, thus we uniformly sample it on the range [0.10-0.60], which are general limits for a lahar, encompassing a wide range of flows, from debris flows (solid concentration $\alpha_s > 0.5$) to hyperconcentrated flows ($\alpha_s = 0.5$–0.1) to muddy streamflows ($\alpha_s < 0.1$) (Vallance and Iverson, 2015; Neglia et al., 2022). Volcanoclastic deposits from Vesuvius are characterised by an effective porosity ($\alpha_d$) that has been estimated as 0.37±0.10 (Di Vito et al, this issue). Thus, $\alpha_d$ is sampled from a Gaussian distribution with mean 0.37 and standard deviation equal to 0.03, so that 99% of the sampled values are within the above estimate.

To identify the possible initial area of lahar formation in a given catchment, we first select the grid points (Figure 1a,b) falling within the catchment. Then, we rely on three empirical assumptions based on field evidence (Bisson et al, 2014): (*i*) only grid cells that are "steep enough", i.e., having a slope larger than a 20-30 degrees, can be the site of remobilization, (*ii*) the steeper a cell, the more likely its deposits will be remobilized, however (*iii*) on very steep cells (slope >40 degrees), deposits tend not to accumulate, thus we assume no deposit available for remobilization on such grid cells.

In order to define the potential initial area of lahar generation accounting for the points (*i*) to (*iii*) above, we assume that remobilization can occur in a grid cell if it is steeper than a given $\delta_{min}$ and less steep than 40 degrees. Then we distinguish between Somma-Vesuvius catchments, where most of the deposits are fine-grained PDC deposits and are thus easier to be remobilized, and Apennines catchments, where deposits are coarser-grained fallout layers, more difficult to be remobilized. As pointed out by Pierson et al (2013) the thickness of fine ash is also important because it can prolong low-infiltration capacity and high runoff rate. To reflect this feature, for each catchment and simulation, the slope threshold $\delta_{min}$ is sampled randomly from a triangular distribution, independently for Somma-Vesuvius and Apennines catchments. For Somma-Vesuvius we considered a lower bound distribution $\delta_{min}$ of [20-30] degrees and for the Apennines [20-35] degrees. In this way, steeper cells are much more likely to be the site of remobilization (to reflect point ii), that can never occur at slopes lower than 20 degrees (to reflect point *i*) and that definitely occurs above 30 and 35 degrees respectively (to reflect the difference in type of deposits). These values are in agreement with Pierson et al (2013).

Having defined the initial solid volumetric fraction and the area of lahar initiation, we can univocally define the initial flow volume, by keeping into consideration some physical constraints given by the availability of solid deposits and of water from pores and from rain in the domain of remobilization. The simulated volumes in each catchment (minimum, maximum, mean and some percentiles) are provided in Table 1. As typical conditions during lahar flows we assume that the deposits are already water-saturated by an extra amount of water from previous rain when the lahar is triggered (e.g., Fiorillo and Wilson, 2004; Di Vito et al., this issue). Following de' Michieli Vitturi et al. (this issue), the thickness of available compacted deposit (i.e. devoid of the water filling its pore), $\bar{h}_s$, can be bounded as:

$$\bar{h}_s \leq min\left\{ h_d\left(1-\alpha_d\right), \frac{\alpha_s\left(1-\alpha_d\right)}{\left(1-\alpha_s-\alpha_d\right)}h_r \right\} \tag{1}$$

being $h_d$ the thickness of the water-saturated deposit, $h_r$ the column of available rainwater, $\alpha_d$ the deposit effective porosity, and $\alpha_s$ the initial solid volumetric fraction. Following the considerations reported in de' Michieli Vitturi et al. (this issue), we set $h_r$ equal to 500 mm as a conservative value corresponding to the maximum 2-day accumulated rainfall over the Somma-Vesuvius area in the last 70 years in the region, although in principle can be sampled from a suitable distribution (in this case assumed a Dirac delta distribution).

**3.2 PVHA workflow and combination of the model simulation output**

After the sampling of the relevant parameters, for our PVHA we follow the step of the workflow illustrated in Figure 4. For each catchment, run $N_s$=100 simulations with the generalised ImexSFlow-2D model (see purple box in Figure 4), each with a different set of initial values for the model parameters. The simulations are run over a sub-domain with a resolution of 50 m × 50 m, cut in order to save computational time to avoid simulating negligible flow thicknesses over very distal grid points from a given catchment.

For each source catchment $i$, we compute the probability (given $N_s$ simulations) of overcoming a given threshold $h_j$ in maximum flow thickness, in a target-grid point $x$, as

$$p_{i,h_j}(x) = \frac{\sum_{k=1}^{N_s} \theta_{ik}}{N_s} \tag{2}$$

where $\theta_{ik}$ is 1 respectively if, in the $k^{th}$ simulation from catchment $i$, the maximum simulated flow thickness in $x$ was larger than $h_j$, or 0 otherwise. The set of thresholds in flow thickness used is composed by 21 values: from 0.1m to 1m at a step of 0.1m, from 1.5m to 4m at a step of 0.5m, from 6m to 10m at a step of 2m, and the last values at 15m and 20m.

Similarly, for each source catchment $i$, we compute the probability of simultaneously overcoming a pair $(h_j, P_l)$ of threshold values in flow thickness and dynamic pressure, in a target-grid point $x$, as

$$p_{i,h_j,P_l}(x) = \frac{\sum_{k=1}^{N_s} \theta_{ik}}{N_s} \tag{3}$$

Here $\theta_{ik}$ is 1 respectively if, in the $k^{th}$ simulation for catchment $i$, the pair $(h_j, P_l)$ has been overcome in $x$ at least once, or 0 otherwise. In this case the set of threshold pairs in flow thickness and dynamic pressure simultaneously overcome is composed of 36 values, consisting of all the possible combinations of the values 0, 0.1, 0.5, 1, 2 and 5 m of thickness, and 0, 0.5, 1, 2, 5, 30 kPa for dynamic pressure. The value equal to 0 in one of the two target variables allows computing the probability maps of overcoming the six thresholds in the other variable. In other words, by selecting a threshold value equal to 0 in one variable and a given threshold $t_i$ in the other variable, we can visualise the probability of a maximum value larger or equal than $t_i$ in the latter variable.

We then combine all the $N_c$=11 catchments together by computing

$$p_{h_j}(x) = 1 - \prod_{i=1}^{N_c} \left(1 - p_{i,h_j}(x)\right) \tag{4}$$

and

$$p_{h_j,P_l}(x) = 1 - \prod_{i=1}^{N_c} \left(1 - p_{i,h_j,P_l}(x)\right) \tag{5}$$

which respectively yield the probability of overcoming the maximum flow thickness $h_j$, and the probability of overcoming simultaneously the pair $(h_j, P_l)$ in flow thickness and dynamic pressure, in the target-grid point $x$ from at least one catchment, given the remobilization of the volcanoclastic deposits from a medium-size eruption at Somma-Vesuvius.

## 4 Results

The ranges in the initial volume of simulated lahars are shown in Table 1. We can see that some catchments are more prone to generate large initial volumes, specifically catchments 1 to 4 (on Somma-Vesuvius) and 7, 8 and 9 (on the Appennines) which have large maximum and/or mean simulated volumes. The latter catchments are characterized by a large extension, whereas the former by the availability of a large thickness of deposits from PDCs. The catchments on Somma-Vesuvius and those on the NorthEastern appennines section (numbers 7 and 8 in our case) were also identified in Tierz et al (2017) as those able to generate larger initial-volume lahars in case of a medium-size eruption. We also notice that in some Appennines catchments (numbers 7 to 11) some simulations do not have significant deposits from tephra fallout to be remobilized (null initial volumes, probably because we simulated deposits from eruptions under wind fields not directed towards those catchments).

Following the approach described in the previous section, for each target-grid point we can build a hazard curve (e.g., Tonini et al., 2015) for the maximum flow thickness from each different catchment (equations 2 and 3) and one from at least one catchment (equations 4 and 5). As an example, we show these results on the grid point located in San Marzano sul Sarno (S in Figure 1a), an inhabited town in the Campanian plain, in figure 5: in panels b-c-d-e we show the hazard curves only relative to the catchments that are able to generate a lahar reaching this target point, and do not show the hazard curves for the other catchments, whereas in panel f we show the hazard curve from any catchment. These hazard curves can be cut at different thresholds in exceedance probability or thickness thresholds, and mapped on the whole target domain, to obtain hazard or probability maps respectively, such as those in figures 6a-k (at 5% exceedance probability) and 7a-k (for a flow maximum thickness of 1 m).

In figures 6l and 7l we show the same hazard and probability maps for at least one lahar from any catchment, deriving from equation 5.

Similarly to hazard curves, we can think of a "hazard surface" in terms of pairs of threshold values, in flow thickness and dynamic pressure, simultaneously overcome at every grid point. These hazard surfaces aim at an easier visualisation for impact assessment: in reality, in order to evaluate the impact of a lahar on the built environment, it is the joint effect of these two parameters that gives a more relevant measure of the impact.

An example for a Vesuvian location (the Casilli train station, point C) is shown in figure 8, from specific catchments (panels b and c) and from any catchment (panel d). This location has been selected as it is impacted by more than one catchment, which is not common on the studied domain. By cutting these surfaces at specific pairs of values for the flow thickness and flow dynamic pressure, we achieve maps showing the probability of simultaneously overcoming the pair of threshold values selected.

In Figure 9 we show the example for 0.5 m of flow thickness and 1 kPa in flow dynamic pressure, from each different catchment (panels a-k) and from any catchment, deriving from catchment (panel l).

Typically, the output of probabilistic hazard studies are a portfolio of scientific products (different hazard and/or probability maps at different thresholds in exceedance probability and/or intensity, respectively, corresponding to different average return times). In this respect, the present study is no different, and in the Appendice A we provide the whole set (apart from those already given in Figs. 6, 7 and 9) of:

- probability maps for the maximum flow thickness (respectively at the other 20 thresholds in thickness considered except the one in Fig. 7 -Figs. SM7 to SM26),
- hazard maps for the maximum flow thickness at the exceedance probability thresholds of 1%, 2%, 10%, 50% and 90% -Figs. SM27 to SM31, and
- probability maps for the simultaneous overcoming of flow thickness and dynamic pressure threshold pairs (at the other 35 threshold pairs considered except the one in Fig. 9, Figs. SM32 to SM65).

The visual inspection of the portfolio of probabilistic maps for lahar maximum thickness shows that, in case of a reference size eruption at Vesuvius and heavy rain, flows of maximum thickness of half a metre (e.g., Figure SM11) are to be expected in the SouthWestern, NorthWestern, Northern, NorthEastern and Eastern sectors around Vesuvius, having conditional probabilities very close to 1 even in valleys down to 10-15 km from the volcano summit. On the SouthWestern sector they would likely reach the shoreline, in agreement with Tierz et al (2017), with decimetric maximum thickness. In such sectors, the hazard maps (e.g., Figure SM30) tell us that a flow of metric maximum thickness has 50% chances to be exceeded, in case of a primary lahar from the reference eruption. In the areas threatened by Apennine catchments, the highest conditional probability for such scale in maximum flow thickness is found in valleys, especially in the Vallo di Lauro, Sarno and Castellammare di Stabia areas, with values up to 50%. This is in agreement with the findings by Tierz et al (2017), who found metric flow depths when considering lahars triggered from Vallo di Lauro and Avella catchments.

Flows of maximum thickness of the order of 1-2 m (e.g., Figures 7 and SM17) have conditional probability close to 1 only in the bottom of valleys from NorthWest to SouthEast (clockwise) sectors around Somma-Vesuvio, up to 5 km approximately. In the other Vesuvian valleys around the volcano they have 10-50% conditional probabilities, whereas in the Apennine valleys the maximum conditional probability is about 30% for this order of flow thickness, again in Vallo di Lauro, Sarno and Castellammare di Stabia areas. Flows of maximum thickness of the order of 5 m or more are conditionally unlikely (probability less than 10%) and confined to valley bottoms, at least at the resolution used in this study.

When accounting for the simultaneous overcoming of conjoint thresholds in flow thickness and dynamic pressure, we can see that for the Vesuvian area the maximum conditional probability for flows of 0.5-1 m and 0.5-5 kPa are found downhill in valley bottoms (e.g., Figures 9, SM45 to SM47, SM50 to SM53), being the highest values located well below 500 m altitude: these areas are densely populated and built. In these areas, these ranges of conjoint thresholds have conditional probability close to 1 in the bottom of the valleys. High dynamic pressure values (e.g. 5 kPa or larger, e.g., Figures SM41-SM42-SM47-

SM48-SM53-SM54) are overcome only in steep slopes, where presumably the flow speeds up. Their conditional probability is quite low in Apennine steep flanks, where such dynamic pressure thresholds are anyway overcome with very thin flows

(of the order of 0.1 m, Figure SM42). In steep flanks of Vesuvian catchments 1 to 4, high dynamic pressure flows (5kPa or larger) are more probable than in the Apennines, still associated with flows of 0.1-0.5 m (Figures SM42 and SM47). In Di Vito et al (this issue), a reverse engineering approach is used to invert the occurrence of external clasts (bricks, walls, limestone fragments, etc…) into the volcaniclastic deposits to estimate the local flow dynamic pressure, velocity and thickness. For flows with an estimated thickness of 0.5-1 m, the most characteristic range of dynamic pressure is 4-8 kPa,

corresponding to a representative range of velocity of 2-4 m/s. This means that the higher the velocity (and dynamic pressure) the higher the capability of the flow to entrain accidental clasts (or damage infrastructures). Comparing such estimates with the probabilistic maps from the present study, we see that the probability to overcome these combinations of thresholds in thickness and dynamic pressure (0.5-1 m; 2-30 kPa, Figures SM46 to SM48, SM52 to SM54) is statistically significant, being larger than 5% in several grid points. In particular, the pairs of lower thresholds (e.g., 0.5m and 2kPa, Fig.

SM46) have a probability larger than 5% in steep valleys throughout the domain and in some locations at the mouth of the narrowest and steepest valleys (e.g., the Vesuvian ones or in Vallo di Lauro). The pairs of higher thresholds (e.g., 1m and 5kPa, Fig. SM53) have a probability larger than 5% only in the steepest valleys on Vesuvius slopes. As a concluding remark, we can state that the estimated combined values of flow thickness and dynamic pressure from field data in Di Vito et al (this issue) are well captured by our probabilistic maps, and do not represent outliers or unlikely values with respect to them.

As mentioned above, the probability maps shown in Figures SM32 to SM37 (where the threshold in maximum flow thickness is 0m) in practice show the probability of flows with maximum dynamic pressure overcoming the 6 thresholds considered (see section 3.2). It is important to highlight that where high values of dynamic pressure are overcome, it may be due to simulated flows of negligible thickness: this is the reason why we decided to consider the simultaneous overcoming of non-0 thresholds in thickness and dynamic pressure (e.g., in Figure 9 and SM38 to SM65). We think that this type of

information could be of great importance when incorporating probabilistic hazard into impact and quantitative risk assessment (e.g., Zuccaro et al, 2008; Zuccaro and De Gregorio, 2013).

Comparing the different catchments, the catchments threatening the largest areas (e.g., where the area invaded by at least 10 cm thick flows with exceedance probability of 10%, Figure SM29) are the Apennine sectors 7 (Vallo di Avella), 8 (Vallo di Lauro) and 9 (upward the towns of Nocera Inferiore, Gragnano and Castellammare di Stabia, a very dense inhabited area).

Overall, the Vesuvian sectors 2 (North), 3 (NorthEast) and 4 (East) show the largest maximum expected thickness in hazard maps, given an exceedance probability (e.g., Figures 6, SM27 to SM31), and the largest probability of given maximum thickness in probability maps (e.g., Figures 7, SM7 to SM26). In few words, the smaller conditional hazard ubiquitously found in the Apennine areas, compared to the Vesuvian ones, is due to the smaller amount of available sediments to be remobilized (only from medio-distal fallout, whereas on Somma-Vesuvio both proximal fallout and PDC deposits are

available) and to their coarser grain size. In our simulation strategy scheme, the latter feature has been assumed as a higher resistance to remobilization.

In a few points (such as S and C, respectively shown in Figs. 5 and 8) the hazard is significantly due to more than one catchment, but this appears to be an exception: overall, most of the inspected domain is threatened by one catchment.

## 5 Discussion and conclusions

The present study aims at providing a probabilistic assessment of lahar hazard conditional to the occurrence of a medium size reference eruption from Somma-Vesuvius. For the first time, such an hazard assessment has considered several sources of uncertainties previously overlooked, such as uncertainties on the initial volumes and initial detach areas of lahars, and effects of erosion and deposition processes. This has been possible thanks to both the model formulation (de' Michieli Vitturi et al., this issue) and the availability of relatively fast computational resources at INGV. Both factors enabled the simulation,

in a reasonable time, of 100 different scenarios from each of the 11 catchments examined here, totalling to a larger-than-ever number of simulations for a lahar hazard assessment.

Where possible, constraints on the range of parameters values, or constraints on the probabilistic distribution describing relevant parameters, were obtained by comparison with field data (for example, the GSD or the sediment porosity $\alpha_d$). For other parameters, only loose constraints were possible, and a maximum-ignorance uniform probability distribution has been

used to describe such cases.

Under the hazard/risk separation principle (Jordan et al, 2014, Marzocchi et al, 2021), we remark that this is the role of volcanologists, that is to quantify uncertain scientific information in a way that can be used to mitigate risk.

The present study does not tackle the variability related to the different possible eruption sizes at Somma-Vesuvius, as we have focussed on the medium-size eruption only (Sandri et al, 2016), and we provide the quantification of lahar hazard in

case of such en event. Future research will be devoted to extend the present analysis to other eruptive sizes, especially to larger ones (i.e., Plinian events similar to Pompeii 79CE or Avellino and Mercato eruptions, see Gurioli et al, 2010), whose lahar hazard potential is significant (Tierz et al, 2017).

However, there is a number of papers in the literature quantifying the probability of a Somma-Vesuvius eruption of medium size similar to the one accounted for in this study (e.g., Marzocchi et al, 2004 and Selva et al, 2022). According to Selva et al

(2022), an estimate of the probability of at least one eruption from Somma-Vesuvius, given the low-activity current period, is about 34% in 50 years (Figure 6 in Selva et al, 2022), and the conditional probability of a medium-size event (i.e., a VEI=4) is about 20% (Figure S7 in Selva et al, 2022). Consequently, a gross estimate of the probability of an eruption of

medium-size at Vesuvius is about 7% in 50 years. To our knowledge, the probability of syn-eruptive lahar generation, given an eruption of medium-size, has never been quantified. It may be broadly estimated on the basis of the frequency of lahar

triggering observed at analogue volcanoes (Tierz et al, 2019) with similar climate. Such development is beyond the goals of this study and is foreseen as a future research.

Finally we remark that, in the present study, two limitations are related to (*i*) the number of simulations performed from each catchment (100), and (*ii*) the assumption of available rainfall, that we have fixed at 50 cm. As for the former limitation, it is mainly dictated by the availability of computational resources. Such a number has been chosen to carry out the simulations

in a reasonable time, and it may be overcome in the future as more performing codes and resources are available. At the same time, the exploration of lahar hazard in case of other possible eruptive size classes from Somma-Vesuvius (Sandri et al, 2016) could be examined in future developments of this work. As for the latter limitation, we have also assessed that the possible rain due to condensation of magma-exsolved water vapour in the umbrella region, that we have roughly estimated in an extra 5 to 10 cm of water, does not significantly increase such an upper limit. We analysed the data from the rain gauge

operating at the historical building of the Osservatorio Vesuviano on Mt Vesuvius since 1940 (Ricciardi et al., 2007), finding that this amount represents approximately the maximum recorded accumulated rain over 2 days since about 1950 in the Campanian region. This value is similar to the maximum rainfall among the episodes of lahars reported by Fiorillo & Wilson (2004). In this perspective, this limit has been taken as conservative. In fact, it implies that the simulated lahars have larger initial volumes, since more water is available, compared to cases with a smaller amount of rainfall. However, due to climate

change, the 2 day intensity of rain may be larger over this area in the coming decades. Furthermore, the comparison of the 2 days accumulated rainfall at Vesuvius with the occurrence of lahar cases in Campania shows very little correlation in time (Cantelli, 2021). Following these considerations, we acknowledge that relaxing this assumption, on the one hand, would allow simulating more frequent and smaller size lahars, that appear to have occurred in the last 50 years even with smaller rainfall intensity (e.g., the Sarno event in 1998), and, on the other hand, would allow accounting for potentially larger rainfall

intensities expected in the future by the ongoing climate change, subject of a future work.

**Supplementary Material**

See attached PDF

**Author Contribution**

*Conceptualization*: L. Sandri, A. Costa, M. de' Michieli Vitturi, M.A. Di Vito
*Data curation:* A. Costa, L.Sandri, M. de' Michieli Vitturi, M.A. Di Vito. D. M. Doronzo, I. Rucco, M. Bisson, R. Gianardi, R. Sulpizio, S. De Vita

*Formal analysis:* L. Sandri, A. Costa, M. de' Michieli Vitturi

*Funding acquisition:* M.A. Di Vito, A. Costa, L.Sandri

*Methodology*: L. Sandri,  A. Costa, M. de' Michieli Vitturi

*Manuscript curation:* L. Sandri,  A. Costa

**Competing interests**

The contact author has declared that none of the authors has any competing interests


**Acknowledgments**

DPC-INGV Allegato BThis work benefited of the 2012–2021 agreement between Istituto Nazionale di Geofisica e Vulcanologia (INGV) and the Italian Presidenza del Consiglio dei Ministri, Dipartimento della Protezione Civile (DPC),

Convenzione B2. The paper does not necessarily represent DPC official opinion and policies.

We thank the colleagues Daniela De Gregorio, Stefano Nardone, and Giulio Zuccaro from PLINIVS Centre at Universita' di Napoli Federico II (Naples, Italy) for the helpful discussions on lahar impact parameters, and Ylenia Cantelli.

We also wish to thank the invaluable work of two anonymous reviewers and of the Editor Virginie Pinel, that improved the quality of the paper.

**References**Auker M.R., Sparks R.S.J., Siebert L., Crosweller H.S., Ewert J. (2013) A statistical analysis of the global historical volcanic fatalities record. *J Appl. Volcanol.* 2, 2 (2013). https://doi.org/10.1186/2191-5040-2-2

Bisson M, Spinetti C, Sulpizio R (2014) Volcaniclastic flow hazard zonation in the Sub-Apennine Vesuvian area using GIS and remote sensing, Geosphere 10 (6), 1419-1431

Cantelli Y (2021) Analisi statistica degli eventi di precipitazione associati alla generazione di lahar nell'area del Vesuvio, Master Thesis, Alma Mater Studiorum - Università di Bologna, in Italian


Cioni R, Bertagnini A, Santacroce R, Andronico D (2008) Explosive activity and eruption scenarios at Somma-Vesuvius (Italy): towards a new classification scheme. J Volcanol Geotherm Res 178:331– 346. doi:10.1016/j.jvolgeores.2008.04.024

Costa A, Dell'Erba F, Di Vito MA, Isaia R, Macedonio G, Orsi G, Pfeiffer T (2009) Tephra fallout hazard assessment at the

Campi Flegrei caldera (Italy). Bull Volcanol 71, 259–273

de' Michieli Vitturi M., Costa A., Di Vito M.A., Sandri L., Doronzo D. (this issue) Lahar events in the last 2,000 years from Vesuvius eruptions. Part 2: Formulation and validation of a computational model based on a shallow layer approach, Solid Earth

Di Vito M.A., Rucco I., de Vita S., Doronzo D.M., Bisson M., de' Michieli Vitturi M., Rosi M., Sandri L., Zanchetta G., Zanella E., Costa A. (this issue) Lahar events in the last 2,000 years from Vesuvius eruptions. Part 1: Distribution and impact on densely-inhabited territory estimated from field data analysis, Solid Earth

Fiorillo F., Wilson R. C. (2004). Rainfall induced debris flows in pyroclastic deposits, Campania (southern Italy). Engineering Geology 75 (2004) 263–289

Gattuso A., Bonadonna C., Frischknecht C. et al. (2021) Lahar risk assessment from source identification to potential impact analysis: the case of Vulcano Island, Italy. J Appl. Volcanol. 10, 9. https://doi.org/10.1186/s13617-021-00107-6

Gurioli, L., Sulpizio, R., Cioni, R., Sbrana, A., Santacroce, R., Luperini, W., & Andronico, D. (2010). Pyroclastic flow hazard assessment at Somma–Vesuvius based on the geological record. Bulletin of Volcanology, 72(9), 1021-1038. https://doi.org/10.1007/s00445-010-0379-2

Jenkins, S. F., Biass, S., Williams, G. T., Hayes, J. L., Tennant, E., Yang, Q., Burgos, V., Meredith, E. S., Lerner, G. A., Syarifuddin, M. & Verolino, A. (2022). Evaluating and ranking Southeast Asia's exposure to explosive volcanic hazards. Natural Hazards and Earth System Sciences, 22(4), 1233-1265. https://dx.doi.org/10.5194/nhess-22-1233-2022

Jordan T.H., Marzocchi W., Michael A., Gerstenberger M. (2014) Operational earthquake forecasting can enhance earthquake preparedness. Seismol. Res. Lett. 85, 955–959

Macedonio G, Costa A, Folch A (2008) Ash fallout scenarios at Vesuvius: Numerical simulations and implications for hazard assessment. J Volcanol Geotherm Res 178, 366–377

Manville V. (2004). Palaeohydraulic analysis of the 1953 Tangiwai lahar: New Zealand's worst volcanic disaster. Acta Vulcanol., 16(1–2), 137–151

Marzocchi W., Papale P., Sandri L., Selva J. (2021) Reducing the volcanic risk in the frame of the hazard/risk separation
principle, in Forecasting and Planning for Volcanic Hazards, Risks, and Disasters, Editors: Schroeder JF, Papale P, Volume 2, 978-0-12-818082-2, https://doi.org/10.1016/B978-0-12-818082-2.00014-7

Massaro S., Stocchi M., Martínez Montesinos B., Sandri L., Selva J., Sulpizio R., Giaccio B., Moscatelli M., Peronace E., Nocentini M., Isaia R., Titos Luzón M., Dellino P., Naso G., Costa A. (2023) Assessing long-term tephra fallout hazard in Southern Italy from Neapolitan volcanoes, Natural Hazards and Earth System Science, https://doi.org/10.5194/nhess-2023-3

Mead S.R., Magill C.R. (2017) Probabilistic hazard modelling of rain-triggered lahars. J Appl. Volcanol. 6, 8. https://doi.org/10.1186/s13617-017-0060-y

Neglia, F., Dioguardi, F., Sulpizio, R., Ocone, R., & Sarocchi, D. (2022). Computational fluid dynamic simulations of granular flows: Insights on the flow-wall interaction dynamics. International Journal of Multiphase Flow, 157, 104281. http://dx.doi.org/10.1016/j.ijmultiphaseflow.2022.104281

Neri A, Aspinall WP, Cioni R, Bertagnini A, Baxter PJ, Zuccaro G, Andronico D, Barsotti S, Cole PD, Esposti Ongaro T, Hincks TK, Macedonio G, Papale P, Rosi M, Santacroce R, Woo G (2008) Developing an event tree for probabilistic hazard and risk assessment at Vesuvius. J Volcanol Geotherm Res 178:397–415. doi:10.1016/j. Jvolgeores.2008.05.014

Neri, A., Bevilacqua, A., Esposti Ongaro, T., Isaia, R., Aspinall, W. P., Bisson, M., Flandoli, F., Baxter, P. J., Bertagnini, A., Iannuzzi, E., Orsucci, S., Orsucci, S., Pistolesi, M., Rosi, M., Vitale, S. (2015). Quantifying volcanic hazard at Campi
Flegrei caldera (Italy) with uncertainty assessment: 2. Pyroclastic density current invasion maps. J. Geophys. Res. Solid Earth 120, 2330–2349

Parra E., Cepeda H. (1990). Volcanic hazard maps of the Nevado del Ruiz Volcano, Colombia. Journal of Volcanology and Geothermal Research, 42: 117-127.

Pierson T.C., Janda R.J., Thouret J.C., Borrero C.A. (1990). Perturbation and melting of snow and ice by the 13 november
1985 eruption of Nevado del Ruiz, Colombia, and consequent mobilization, glow and deposition of lahars. Journal of Volcanology and Geothermal Research, 41: 17-66.

Pierson T.C., Daag A.S.,Delos Reyes P.J., Regalado M.T., Solidum R.U., Tubianosa B.S. (1996). Flow and Deposition of Posteruption Hot Lahars on the East Side of Mount Pinatubo, July-October 1991, in: FIRE AND MUD: ERUPTIONS AND LAHARS OF MOUNT PINATUBO, PHILIPPINES, Edited by Newhall C.G. and Punongbayang R.S., Philippine Institute
of Volcanology and Seismology, Quezon City, University of Washington Press, Seattle and London

Pierson T. C., Major J. J., Amigo Á., Moreno H. (2013). Acute sedimentation response to rainfall following the explosive phase of the 2008-2009 eruption of Chaitén volcano, Chile. Bull. Volcanol. 75, 1–17. doi: 10.1007/s00445-013-0723-4

Pizzimenti L., Tadini A.,Gianardi R., Spinetti C., Bisson M., Brunori C.A. (2016)." Digital Elevation Models derived by ALS data: Sorrentina Peninsula test areas". Rapporto Tecnico INGV – N° 361. ISSN 2039 – 7941

Ricciardi, G.P., Siniscalchi, V., Cecere, G., and Macedonio, G., (2007), Meteorologia Vesuviana dal 1864 al 2001 (Dataset)

Rodolfo K.S., Umbal J.V., Alonso R.A., Remotigue C.T., Paladio-Melosantos M.L., Salvador J.H.G., Evangelista D., Miller Y. (1996) Two Years of Lahars on the Western Flank of Mount Pinatubo: Initiation, Flow Processes, Deposits, and Attendant Geomorphic and Hydraulic Changes, in: FIRE AND MUD: ERUPTIONS AND LAHARS OF MOUNT PINATUBO, PHILIPPINES, Edited by Newhall C.G. and Punongbayang R.S., Philippine Institute of Volcanology and
Seismology, Quezon City, University of Washington Press, Seattle and London

Rosi M., Principe C., Vecci R. (1993) The 1631 Vesuvius eruption. A reconstruction based on historical and stratigraphical data, J Volcanol Geotherm Res, 58, 1–4, 151-182, https://doi.org/10.1016/0377-0273(93)90106-2

Sandri L., Costa A., Selva J., Tonini R., Macedonio G., Folch A., Sulpizio R. (2016) Beyond eruptive scenarios: assessing tephra fallout hazard from Neapolitan volcanoes. Sci Rep 6:24271, doi: 10.1038/srep24271

Sandri L., Tierz, P., Costa, A., Marzocchi, W. (2018). Probabilistic hazard from pyroclastic density currents in the Neapolitan area (Southern Italy), Journal of Geophysical Research: Solid Earth, 123, 3474-3500, doi: 10.1002/2017JB014890

Schilling SP (1998) LAHARZ: GIS programs for automated delineation of lahar hazard zones, U.S. Geological Survey Open-file Report

Selva J, Costa A, Marzocchi W, Sandri L (2010) BET VH: exploring the influence of natural uncertainties on long-term hazard from tephra fallout at Campi Flegrei (Italy). Bull Volcanol 72:717–733. doi:10.1007/s00445-010-0358-7

Selva J, Costa A, De Natale G, Di Vito MA, Isaia R, Macedonio G (2018) Sensitivity test and ensemble hazard assessment for tephra fallout at Campi Flegrei, Italy, J Volcanol Geotherm Res 351, 1-28, DOI:10.1016/j.jvolgeores.2017.11.024

Selva J, Sandri L, Taroni M, Sulpizio R, Tierz P, Costa A (2022) A simple two-state model interprets temporal modulations in eruptive activity and enhances multivolcano hazard quantification, Sci Adv, 8 (44), eabq4415, DOI: 10.1126/sciadv.abq4415

Sulpizio R., Mele D., Dellino P., La Volpe L. (2005) A complex, Subplinian-type eruption from low-viscosity, phonolitic to tephri-phonolitic magma: the AD 472 (Pollena) eruption of Somma-Vesuvius, Italy. *Bull Volcanol* **67**, 743–767 (2005).
https://doi.org/10.1007/s00445-005-0414-x

Sulpizio R., Zanchetta G., Demi F., Di Vito M.A., Pareschi M.T., Santacroce R. (2006) The Holocene syneruptive volcaniclastic debris flows in the Vesuvian area; geological data as a guide for hazard assessment. In: Neogene-Quaternary

continental margin volcanism; a perspective from Mexico, Claus Siebe (editor), Jose Luis Macias (editor) and Gerardo J. Aguirre-Diaz(editor), Special Paper - Geological Society of America (2006) 402: 217-235

Tarquini S., Isola I., Favalli M., Mazzarini F., Bisson M., Pareschi M.T., Boschi E. (2007). "TINITALY/01: a new Triangular Irregular Network of Italy", Annals of Geophysics, 50, 3: 407 – 425

Tierz P., Woodhouse M.J., Phillips J. C., Sandri L., Selva J., Marzocchi W., Odbert H.M. (2017) A framework for probabilistic multi-hazard assessment of rain-triggered lahars using Bayesian Belief Networks, Front Earth Sci, https://doi.org/10.3389/feart.2017.00073

Tierz P., Loughlin S.C., Calder E.S. (2019) VOLCANS: an objective, structured and reproducible method for identifying sets of analogue volcanoes. Bull Volcanol 81, 76. https://doi.org/10.1007/s00445-019-1336-3

Tonini R, Sandri L, Thompson M. (2015) PyBetVH: A Python tool for probabilistic volcanic hazard assessment and for generation of Bayesian hazard curves and maps. Comput Geosci. 79:38–46Umbal J.V., Rodolfo K.S. (1996) The 1991 lahars of southwestern Mount Pinatubo and evolution of the lahar-damned Mapanuepe Lake, in: FIRE AND MUD: ERUPTIONS

AND LAHARS OF MOUNT PINATUBO, PHILIPPINES, Edited by Newhall C.G. and Punongbayang R.S., Philippine Institute of Volcanology and Seismology, Quezon City, University of Washington Press, Seattle and London

Vallance J.W. and Iverson R.M.: Lahars and their deposits, in : The encyclopedia of volcanoes, edited by Haraldur Sigurdsson, Academic Press, 649-664, https://doi.org/10.1016/B978-0-12-385938-9.00037-7, 2015.

Voight B., Calvache M.L., Hall M.L., Monsalve M.L. (2013). Nevado del Ruiz Volcano, Colombia 1985. In: Bobrowsky,

P.T. (eds) Encyclopedia of Natural Hazards. Encyclopedia of Earth Sciences Series. Springer, Dordrecht. https://doi.org/10.1007/978-1-4020-4399-4_253

Wilson G., Wilson T.M., Deligne N.I., Cole J.W. (2014) Volcanic hazard impacts to critical infrastructure: A review, J Volcanol Geotherm Res, 286, 1, 148-182, https://doi.org/10.1016/j.jvolgeores.2014.08.030

Zanchetta G., Sulpizio R., Di Vito M.A. (2004a) The role of volcanic activity and climate in alluvial fan growth at volcanic areas: an example from southern Campania (Italy), Sedimentary Geology 168 (3-4), 249-280

Zanchetta G., Sulpizio R., Pareschi M.T., Leoni F.M., Santacroce R. (2004b) Characteristics of May 5–6, 1998 volcaniclastic debris flows in the Sarno area (Campania, southern Italy): relationships to structural damage and hazard zonatio, Journal of volcanology and geothermal research 133 (1-4), 377-393

Zuccaro G., Cacace F., Spence R.J.S., Baxter P.J. (2008) Impact of explosive eruption scenarios at Vesuvius, J. Volcanol.

Geotherm. Res., 178, 416-453

Zuccaro G. and D. De Gregorio (2013) Time and space dependency in impact damage evaluation of a sub-Plinian eruption at Mount Vesuvius, Nat. Hazards, 68, 1399-1423

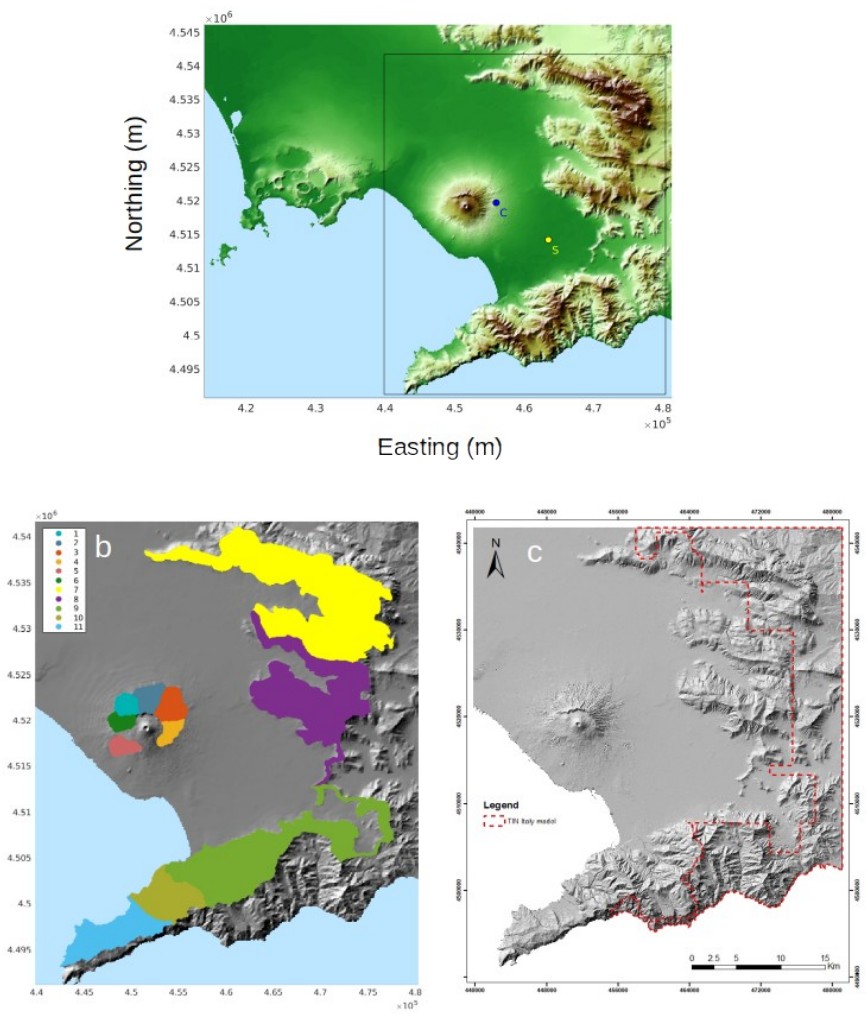

**Figure 1: a**: Campanian Plain highlighting the target area of the present study (black rectangle) covered by a 50 m × 50 m grid; the coloured dots show respectively the location of two illustrative points, that are San Marzano sul Sarno (S) town and Casilli (C) train station, that will be used as examples in the paper. **b**: Shaded topography zoom on the target area highlighting the 11 catchments

considered (1-6 and 7-11 respectively for Vesuvius and Apennines catchments). **c**: the red dashed line encompasses the part of the domain where the TIN Italy DEM has been used.

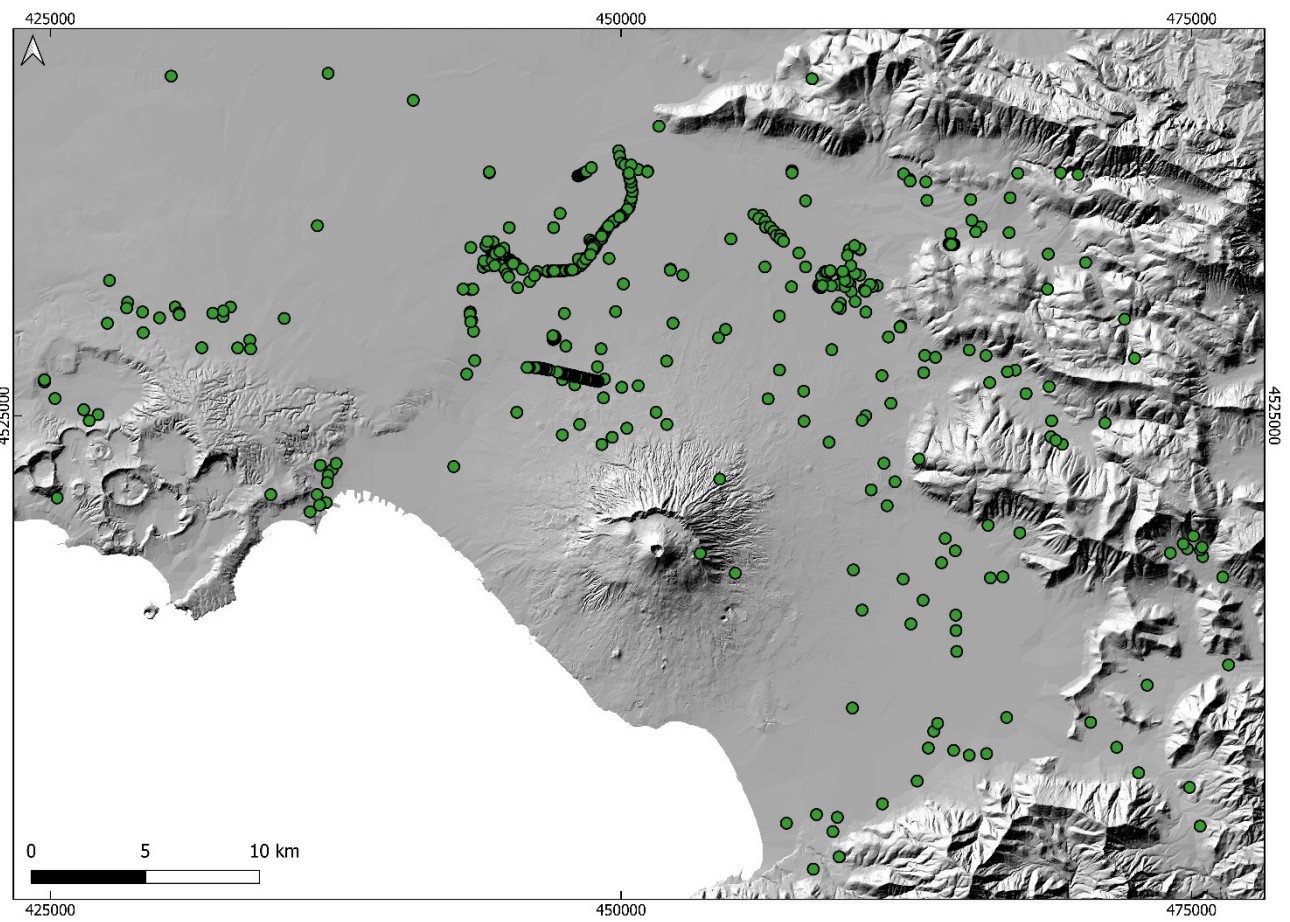

**Figure 2:** The dots highlight the locations of field data analysed in Di Vito et al (2023), where measurements and facies analyses of lahar deposit features were taken or retrieved by an inverse engineering approach.

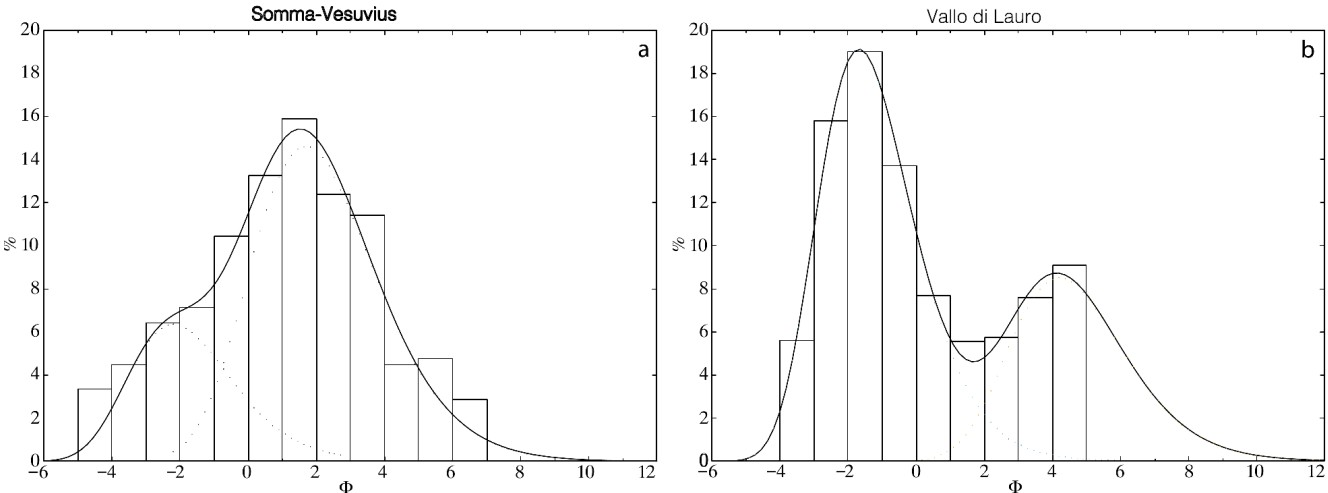

**Figure 3:** Grain Size Distribution used for Vesuvian (a) and Appenine (b) catchments. The former has been derived from local samples from the Somma-Vesuvius catchments, while the latter from samples from Vallo di Lauro (located at the basis of catchment 8 in Fig.1b)

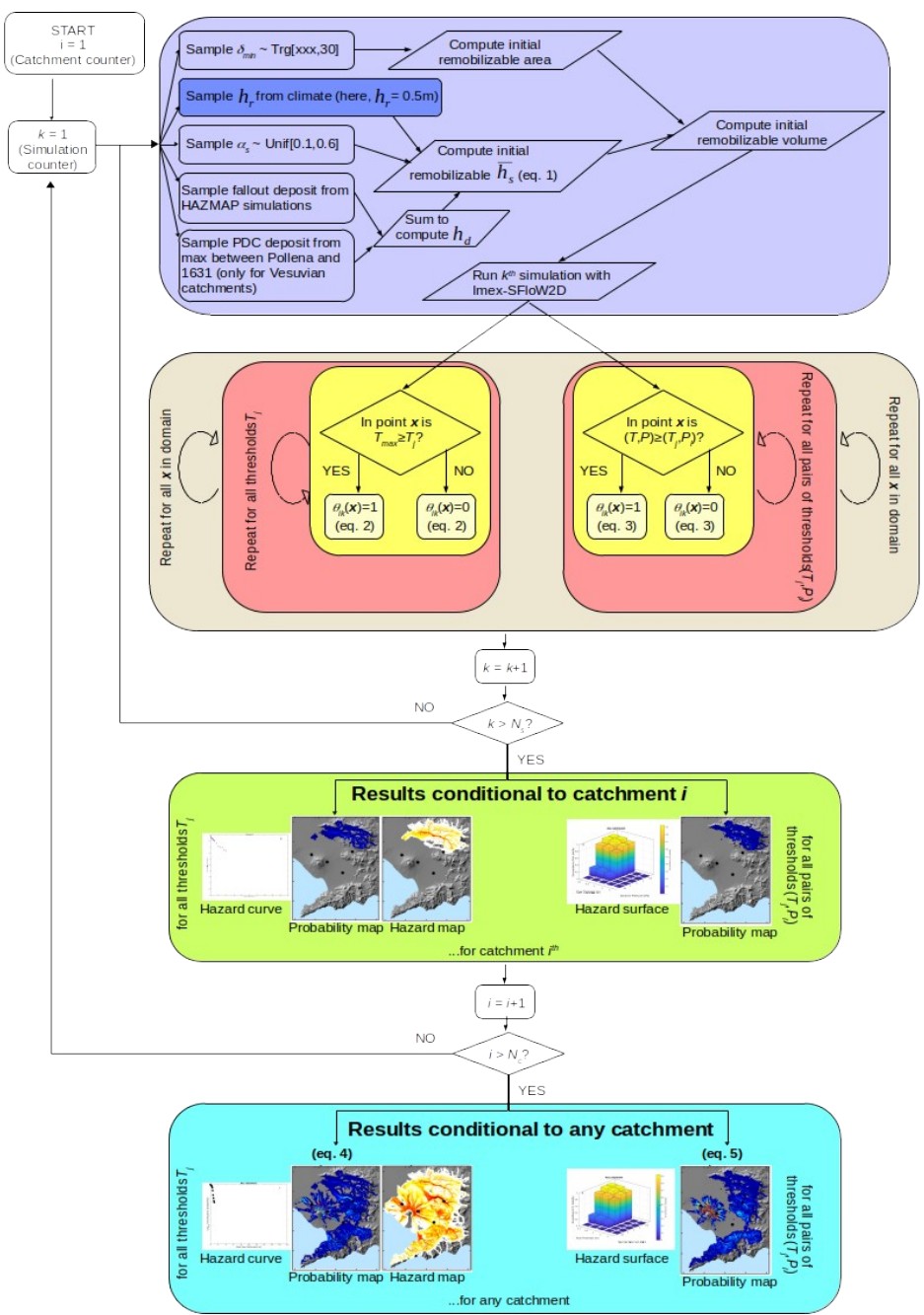

**Figure 4:** Workflow describing the sampling of ESP (and their combination through equation 1) and Imex-SFloW2D simulation (purple box), the postprocessing of the simulation outputs (brown, pink and yellow boxes) to apply equations 2 and 3 to achieve the hazard curves, maps and surfaces, and probability maps for a given catchment (green box), and the further processing through equations 4 and 5 to achieve those for any catchment (cyan box). The number of catchments $Nc$ considered in this application is $Nc$=11, and the number of simulations for each catchment Nsim=100. The darker purple sub-box indicates that, in the present study, we set hr fixed at 0.5 m, but it can ideally be sampled from climatological distributions.

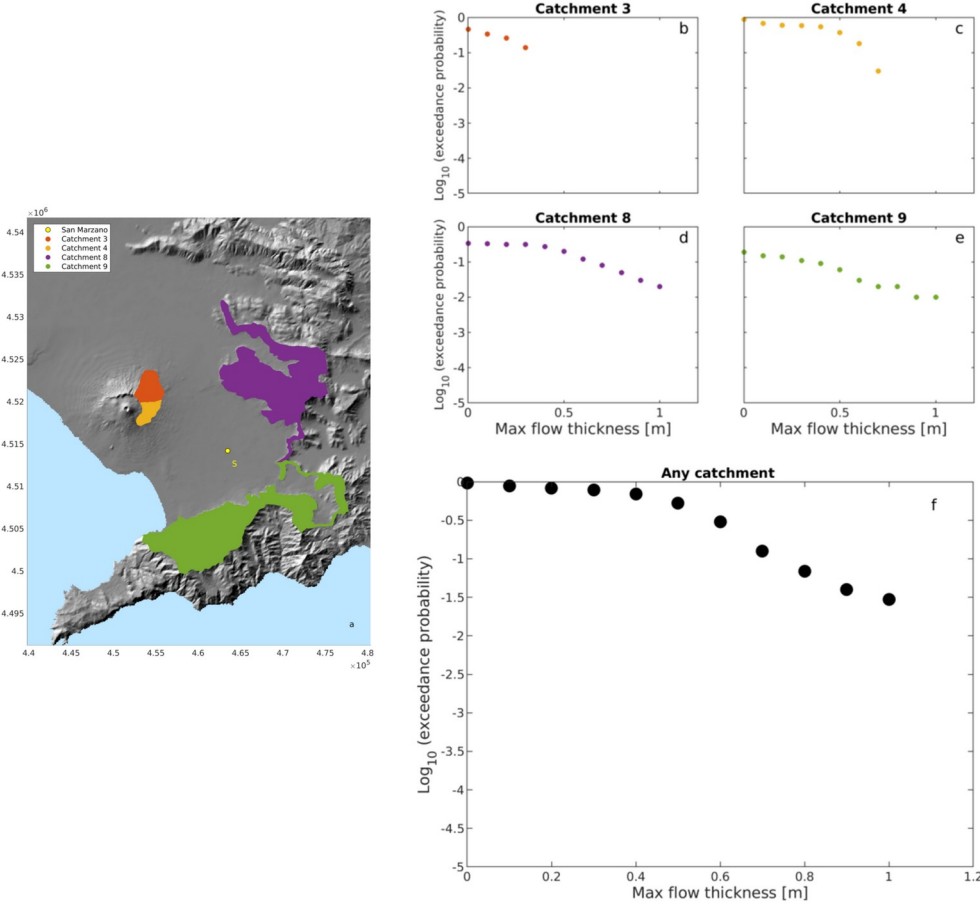

**Figure 5:** Hazard curves for the maximum flow thickness from the catchments 3-4-8-9 (shown in panel a) insisting on San Marzano sul Sarno (point S) (the hazard curves are shown respectively in panels b, c, d, e) and from any catchment (f).

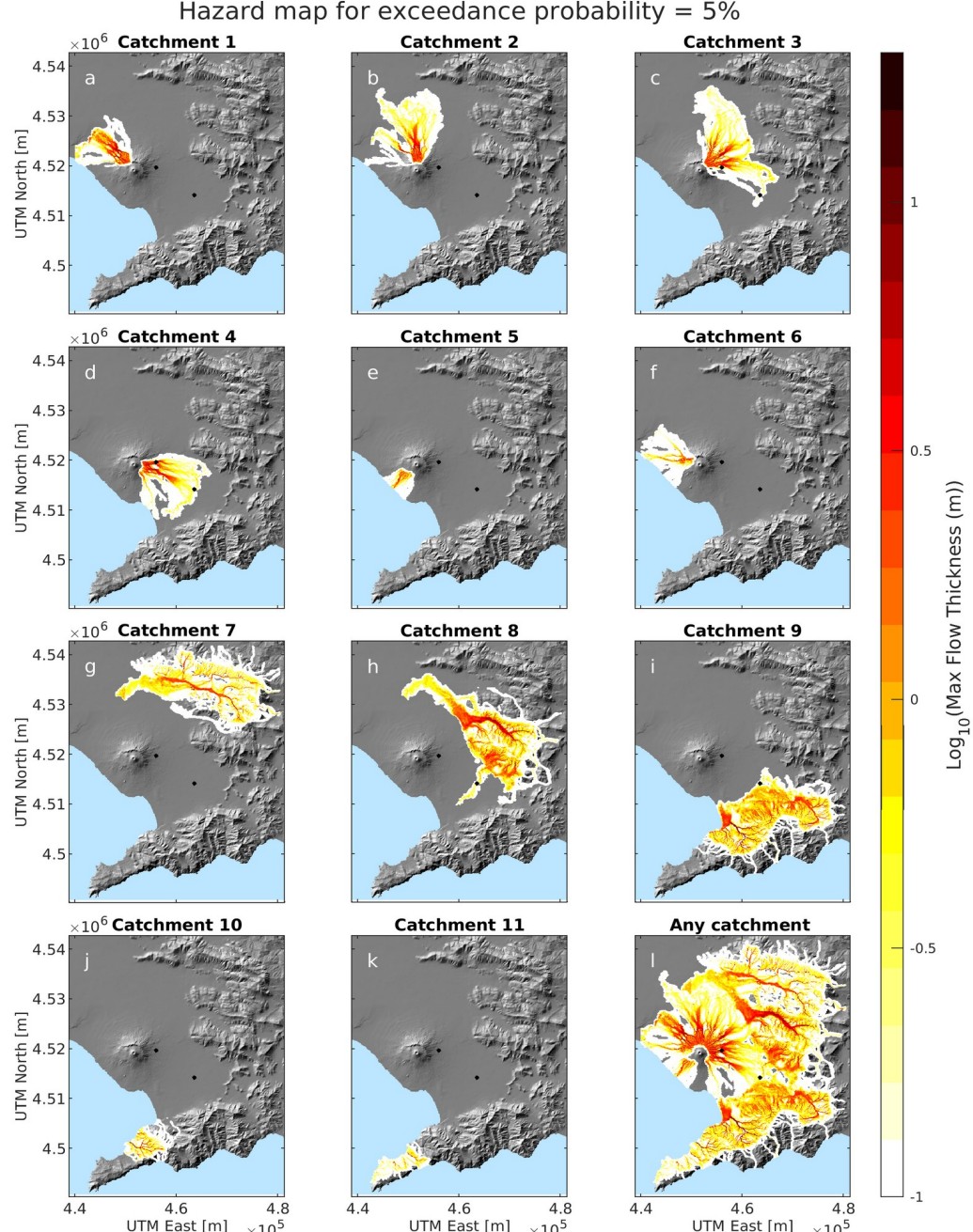

**Figure 6:** Hazard map at the 5% probability level, showing the maximum flow thickness having that probability to be exceeded given a lahar from specific catchments (a-k) and from any catchment (l).

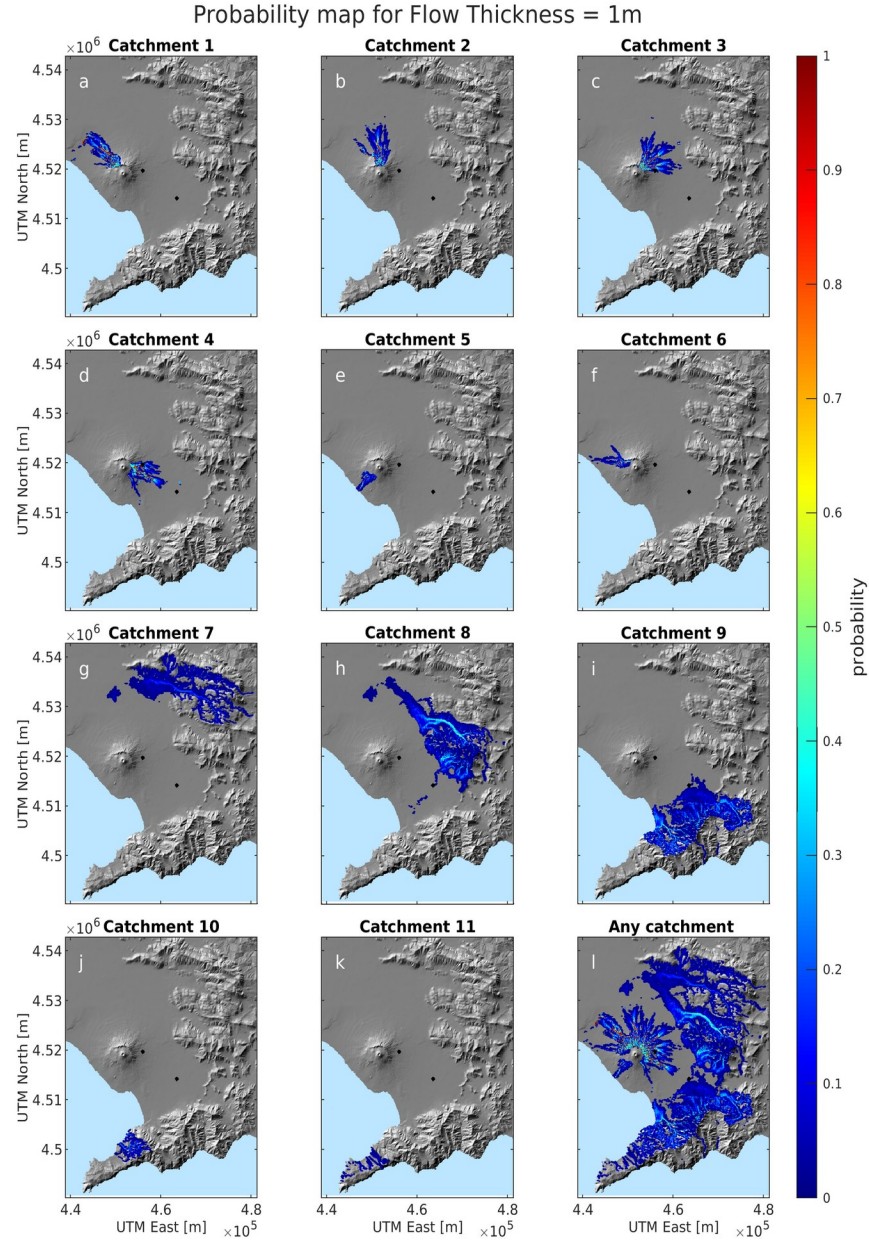

**Figure 7:** Probability map for the maximum flow thickness of 1 m, showing the probability to exceed it given a lahar from specific catchments (a-k) and from any catchment (l).

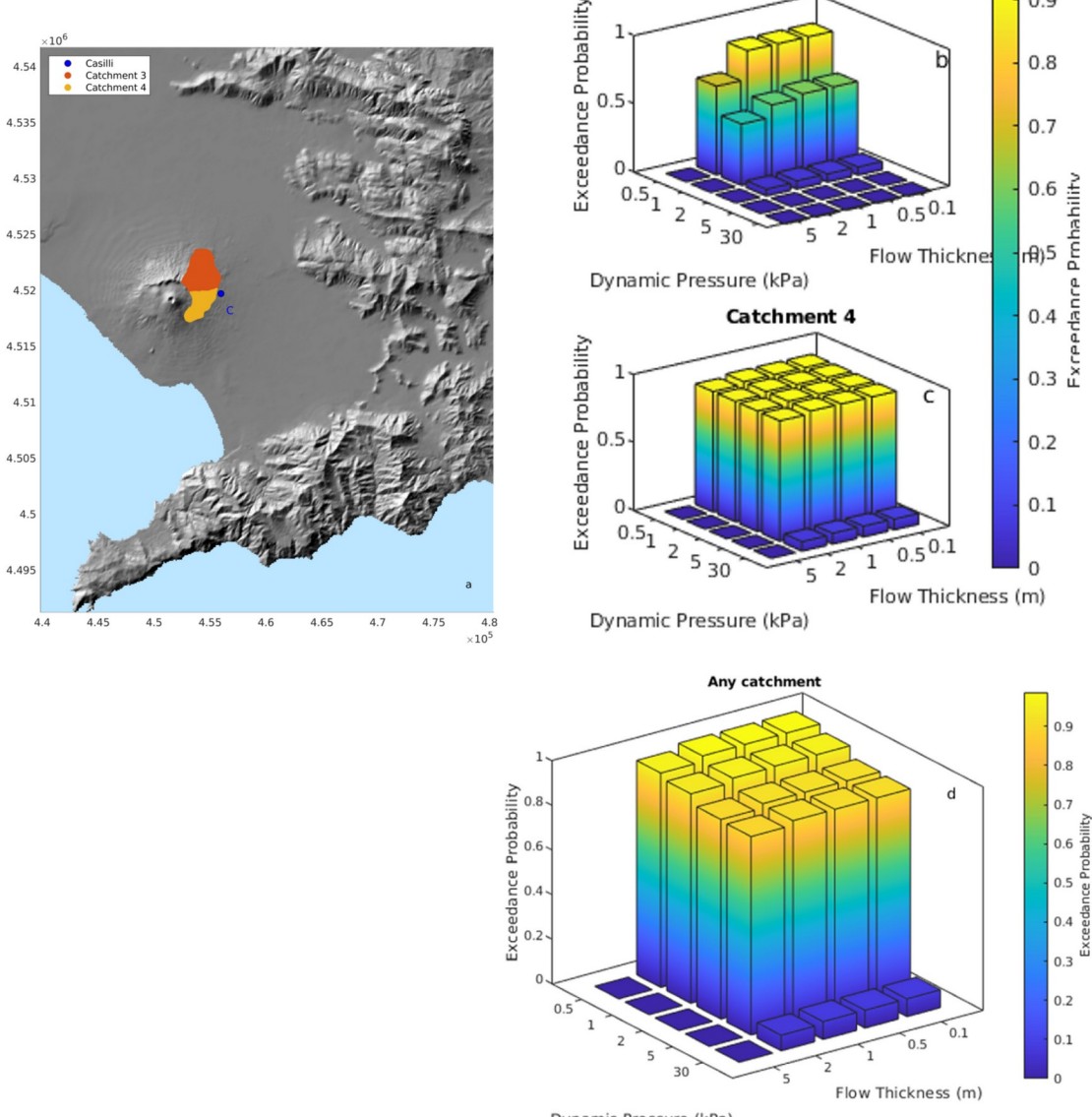

**Figure 8:** Hazard surface for the simultaneous overcoming of different pairs of values in maximum flow thickness and flow dynamic pressure, in the Casilli (C) train station. The panels b and c respectively show the probability to simultaneously exceed the threshold pairs at least once during the flow, given a lahar from catchments 3 and 4 (location of catchments shown in panel a). The panel d shows the hazard surface from all the catchments.

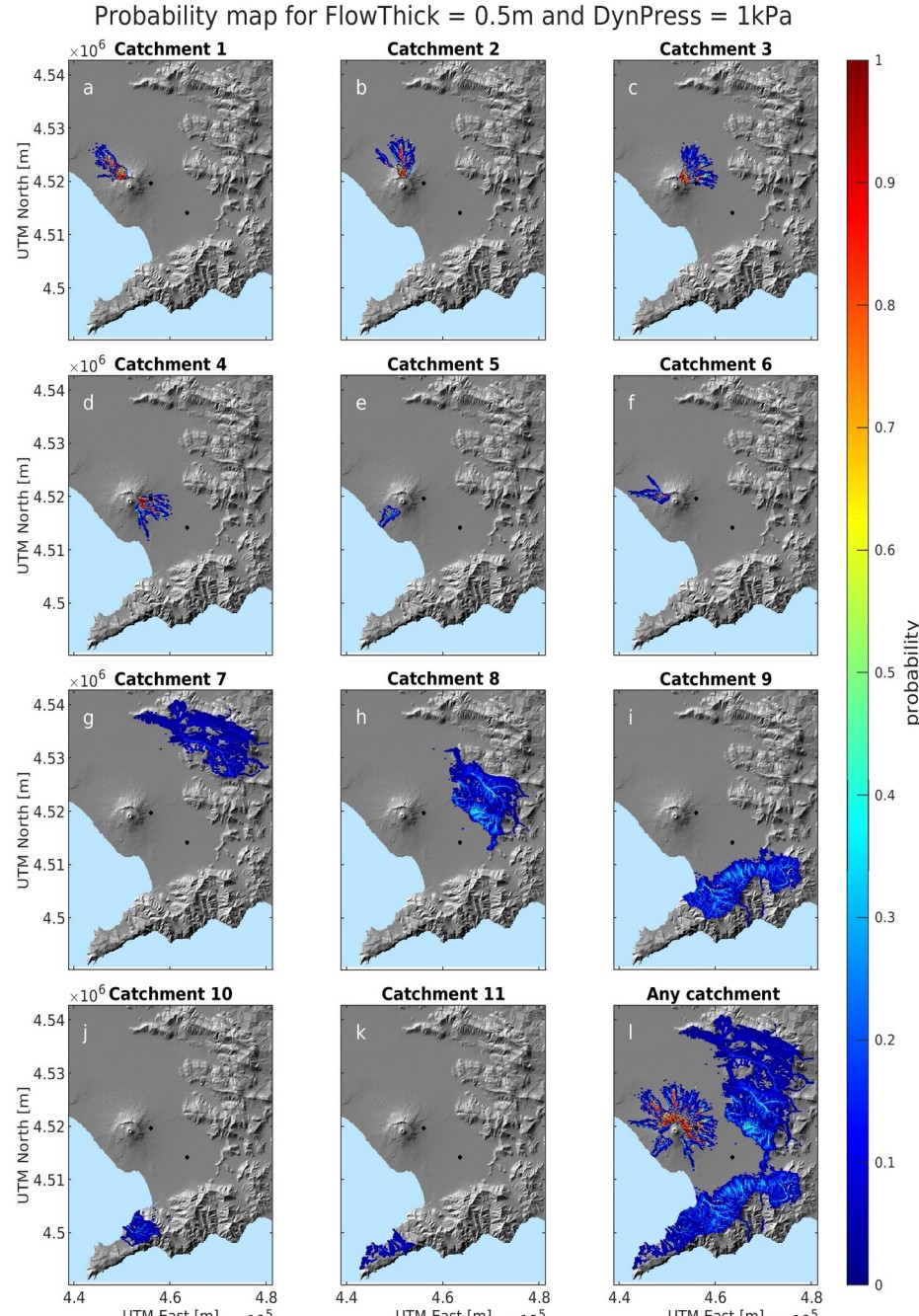

**Figure 9:** Probability map for the simultaneous overcoming of 0.5m in maximum flow thickness and 1kPa in flow dynamic pressure, showing the probability to simultaneously exceed the two thresholds at least once during the flow, given a lahar from specific catchments (a-k) and from any catchment (l).

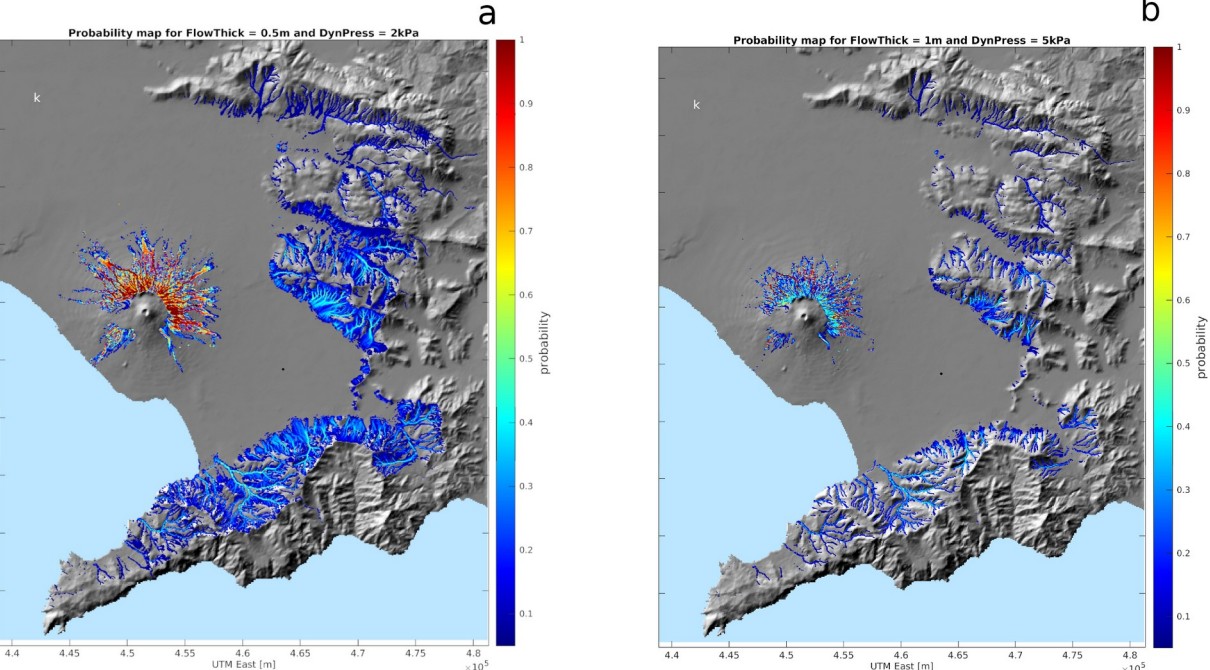

**Figure 10:** Probability maps for the simultaneous overcoming of (a) 0.5m in maximum flow thickness and 2kPa in flow dynamic pressure, and (b) 1m and 5kPa. The maps show only points where the probability to simultaneously exceed the two thresholds at least once during the flow is larger than 5%, given a lahar from any catchment.

| | Volume min | Volume max | 5th percentile | 50th percentile | 95th percentile | mean |
|---|---|---|---|---|---|---|
| Catchment 1 | 1.19E+06 | 6.97E+06 | 1.28E+06 | 2.34E+06 | 5.17E+06 | 2.73E+06 |
| Catchment 2 | 1.18E+06 | 8.71E+06 | 1.31E+06 | 2.53E+06 | 6.04E+06 | 2.97E+06 |
| Catchment 3 | 1.25E+06 | 1.58E+07 | 1.72E+06 | 3.25E+06 | 7.52E+06 | 3.80E+06 |
| Catchment 4 | 8.65E+05 | 8.18E+06 | 1.06E+06 | 1.81E+06 | 4.61E+06 | 2.28E+06 |
| Catchment 5 | 2.37E+05 | 2.64E+06 | 2.88E+05 | 6.17E+05 | 1.89E+06 | 7.37E+05 |
| Catchment 6 | 2.83E+05 | 2.25E+06 | 3.31E+05 | 6.40E+05 | 1.70E+06 | 7.96E+05 |
| Catchment 7 | 2.23E+02 | 1.78E+07 | 0.00E+00 | 1.97E+05 | 8.33E+06 | 1.75E+06 |
| Catchment 8 | 1.58E+02 | 2.95E+07 | 0.00E+00 | 2.50E+06 | 1.97E+07 | 5.11E+06 |
| Catchment 9 | 1.39E+02 | 2.86E+07 | 0.00E+00 | 3.06E+06 | 2.11E+07 | 6.32E+06 |
| Catchment 10 | 8.22E+00 | 8.04E+06 | 0.00E+00 | 1.42E+05 | 4.94E+06 | 9.96E+05 |
| Catchment 11 | 9.05E+00 | 3.34E+06 | 0.00E+00 | 4.43E+03 | 2.06E+06 | 3.01E+05 |

575    Table 1. Values (in m$^3$) of the volume of simulated flows in each catchment. Numbering of catchments is the same as in figure 1b. In the different columns we provide the minimum (non-zero) simulated volume, the maximum, three percentiles (5$^{th}$, 50$^{th}$ and 95$^{th}$) and the mean.