# Peer review of "Lahar events in the last 2,000 years from Vesuvius eruptions. Part 3: Hazard assessment over the Campanian Plain"

_EGUsphere, 2023_

## Author Comment (AC2)

*Dear Editor,*

*please find below a rebuttal letter, where we answer (in blue) point by point to all the comments made by the reviewers and by yourself. We would like to warmly thank the reviewers and you for the careful reading and for the helpful comments which we believe have improved the manuscript.*

*We refer to the tracked-changes version for line numbering.*

*Kind regards,*

*Laura Sandri on behalf of all co-authors*

**RC2**: 'Comment on egusphere-2023-1295', Anonymous Referee #2, 02 Oct 2023

Dear Authors

I have reviewed the paper, "Lahar events in the last 2000 years from Vesuvius eruptions. Part 3: Hazard assessment over the Campanian Plain."

The paper was well written and logically ordered. I enjoyed reading the probabilistic approach to lahar hazards and how the outputs of the analysis have been presented. This paper was difficult to review. The probabilistic model and outputs presented are good and would be of interest to the wider community as well as presenting a very efficient and effective method of presenting lahar hazard analysis. The manuscript though was very short and due to that tended to "gloss over" key aspects of lahar hazard as well as determination of key inputs of the probabilistic hazard analysis. In places this resulted in the paper seeming to be incomplete in understanding lahar hazard. Of note is that the abstract itself seems to read more like a "highlights section" rather than a proper abstract that succinctly reports on the methods and findings of the study.

*We have extended the abstract a bit to put it into a more suitable shape, in particular to frame the present paper (Part 3) as a logical consequence to the other two companions (Part 1 and 2), and to better highlight the importance of a PVHA for lahars simultaneously in terms of flow thickness and dynamic pressure.*

While I also understand that this manuscript is part of a series of publications focussed on the Campanian Plain (Southern Italy) this particular manuscript is focussed on such a localised issue that the manuscript does not provide any indication on how this analysis may be of relevance internationally. Of note is that referencing throughout the manuscript seems very limited. the biggest annoyance in this aspect is the amount of self-citation to unpublished work ("this issue") which makes the paper very hard to review particularly when justifying input data for the probabilistic analysis.

*As the reviewer points out, the amount of self-citations to the companion papers (which are however published in SE Discussion Forum, but we agree are self-citations) is necessary as this paper stems from the findings of those two. So we cannot avoid this kind of self-citations.*

*Concerning the comment on the localised issue, we thought that we already highlighted the relevance of lahar hazard internationally (see Lines 30-32). However, we expanded this part and we added a few examples worldwide with appropriate references (Lines 32-37)*

In general, the manuscript, the lack of referencing, seems to disregard all previous work on lahar sedimentology, lahar trigger mechanisms, understanding of lahar rheology, lahar simulations and hazard analysis in favour of papers included in the same issue from the same authors. Not having that robust background information makes it hard to assess the manuscript and hazard models produced. Essentially most of the input information for the probabilistic analysis is derived from data that is difficult to assess for its applicability to the model presented.

*This is a companion paper, and these topics have been covered in the other two companion papers (lahar sedimentology and lahar trigger mechanisms in the Part 1, understanding of lahar rheology and lahar simulations in Part 2).*

More specific comments are as follows.

- The abstract mentions some representation of how uncertainty on the initial flow conditions is treated or accounted for. I am assuming that is related to the initial flow volumes, materials and rainfall required to trigger a lahar event, but it seems to be hard to find how this was accounted or in the body of the manuscript over than line 194-196 which does not address uncertainty.
  *We are not sure about the line numbering the reviewer is referring to (in our pdf submitted as first version, in lines 194-196 there is no mention to uncertainty on the initial flow conditions). We explained how we account for the uncertainty in such conditions (section 3.1). In any case, to be more clear about this, we have also added a sentence in the Introduction (Lines 109-110).*

- Lines 30-32 identify bulking and debulking or other concepts of variability in the flow, but I do not see how the modelling approach; ImexSflow 2D (not referenced; having to search Github) accounts for some of these. In many respects the simulation strategy does not differ from other Voellmy-Salm based simulations (e.g., Titan 2D, Volcflow, Avaflow) so why the need to recreate these tools with no real improvement to the initial starting and flow conditions or its specific applicability to rain-triggered lahars remobilising material.
  *Effects of erosion and deposition have not been accounted in any of the simulations of lahar emplacement and none of the code cited by the reviewer dealt with such a problem. If the reviewer is aware of any study related with this problem please let us know. However the cited models have in common only the fact that are based on the shallow layer approach.*

- A minor point in lines 33-34 is the listing of a sedimentological characteristics of lahars which may not be factors required for hazard analysis but more relevant for numerical simulation.

Hazard analysis inputs should be focussed on occurrence and magnitude with related hazard impacts characteristics being inundation depth, velocity, dynamic pressure etc.

*We would like to point out that the result of the numerical simulations of lahars made with different initial conditions and from different catchments are the pillars over which our hazard assessment is made. So we do not really understand this comment by the reviewer. Differences in the initial conditions have an effect on the simulation output, and, in turn, on the probabilistic hazard assessment based on the frequency of inundation and related parameters in a given grid point.*

- In lines 38-39, 56 and again in the discussion there seems to be no mention of recent statistical surrogate and emulation approaches (Mead et al., 2023; Tierz et al., 2017; Bayarri et al., 2009, etc.) to account for variability in trigger mechanism and source conditions. Aligned to this is the complete lack of recognition of Mead & Magill (2017) and George et al. (2022) (D-Claw simulations) to provide better constraints over the trigger mechanism of rain triggered lahars (e.g., line 70-) that consider source pore porosity and liquefaction processes.
*We thank the reviewer for this very good observation. We have added the referencing to some of these papers in the revised version (in the introduction, Lines 37-41). We have not stressed the specificity of the methods (statistical surrogates and emulators) but we have cited the applications relevant for lahars (see also the added items in the tracked-change list of references).*

- Several times throughout the paper terms like "a large number of lahar simulations", "a massive set" were used when these should be quantified.
*We have changed the adjectives "large" and "massive" into "considerable" and added the effective number of simulations performed (1100).*

- An issue that needs addressing is the use of past deposit data being analysed to inform model simulation inputs or initial starting conditions (line 90, section 2.3, etc). I struggle to see how the porosity of deposited volcani-clastics which is a mixture of material consisting of source material plus entrained clasts that has been saturated, sorted and probably lithified is representative of the original source material.  Defining the porosity of the source material needs to be better defined.
*This point is fully discussed in the companion paper Part 1 (Di Vito et al, this issue).*

- In terms of the modelling a 50x50 m computational grid seems quite course in relation to the assumptions made on slope and discrimination of catchments and hazard zones.
*In the companion paper Part 2 we run some sensitivity tests on the resolution of the computational grid (from 10 m to 100 m), and we came to the conclusion that 50m x 50m is a good balance between computational feasibility and stability of results.*

- An aspect of the study which could with further explanation which i struggled with was the definition of the hazard over time. As I understand it the hazard analysis is simply related to

flow depth and dynamic pressure from two subplinian eruptions rather than a more complete record of events over the history of the volcanic centre. Should the probability maps and outputs also be prefaced or conditional on the probability of those eruptions occurring again producing a specific volume of available material that could be remobilised.

*In this study we have focused on the hazard from lahar CONDITIONAL to the remobilization of deposits from a medium size eruption from Vesuvius. In particular we considered a sub-plinian eruptive scenario that is the reference size for the Italian Civil Protection. This was clearly stated several times in the manuscript submitted, so we do not understand such a criticism. However we are very aware of this limitation, and in future studies we plan to extend to other eruptive scenarios and conditions, as already mentioned in the Conclusions, and now stressed at Lines 362-366.*
*In any case, there is no direct link with time (we do not provide explicitly the return time of such events, for example). However, there are previous and recent studies assessing the probability of an eruption of reference size in a given time window for Vesuvius. Thus, in the revised manuscript, we have tried to stress this point, by adding the sentence "conditional to the occurrence of an eruption of reference size" everywhere we mention the hazard we assess. Furthermore, in the "Discussion and conclusions" of the revised manuscript (Lines 367-375), we have added the reference to Selva et al (2022) for a more specific quantification of the expected frequency of referenze-size events at Somma-Vesuvius, that is approximately of 7% in 50 years. We have also stressed that, however, the probability of triggering of syn-eruptive lahars has, to our knowledge, never been quantified, and will be subject of future research. As well, the assessment of the hazard from any eruptive size will be the subject of future research, and this is now written explicitly at Lines 362-366, as said above.*

I enjoyed reading the manuscript and I believe the manuscript should be published but it needs to be placed in context with the considerable volume of international literature available on the topic of lahar hazard analysis.

*We thank the reviewer for the careful review.*

**RC1**: 'Comment on egusphere-2023-1295', Anonymous Referee #1, 14 Sep 2023 reply

I read with great interest the paper of Sandri et al. about probabilistic hazard assessment due to lahars on Vesuvius volcano. As mentioned by the authors, lahars are one of the most dangerous volcanic phenomena and this is a welcomed contribution to advance our knowledge on this topic. In general, I think this is a very clear, concise, and useful contribution and I strongly recommend publication. I have some remarks that I think will improve the final product.

*We thank the reviewer for the careful review.*

1. I think it is worth to mention in the introduction some of the codes and publications that have been used in the past to simulate lahars and build hazard maps such as LAHARZ, Laharflow, etc. This can be included in lines 40-45.

   *We have added a sentence on this in the introduction (Lines 37-41).*

2. In the discussion on how to estimate the volume of material removed from slopes, the publication of Pierson et al. 2013 (bull volcanol 75:723) about lahars due to rainfall in Chaiten volcano can be useful to strengthen the criteria used by the authors.

   *We have added reference to this paper when discussing the angles of remobilization we use (section 3.1, Lines 199-200 and Line 205).*

3. I think that in some place, some numbers regarding the range of discharge rate (Q, m3/s) and initial or total volume (V) of the simulated lahars should be given. As most people working on lahars are familiar with and use these measurements, it would be useful to translate the degree of initial erosion, rainfall, etc into Q and V to compare with other volcanoes or lahar cases around the world.

   *We thank the reviewer for this very important point. We have now added a table (Table 1) with the ranges of volume used and some statistics (mean and percentiles) and discussed it shortly in the Results (Lines 249-257).*

4. As the authors claim that this is a novel methodology for probabilistic lahar hazard assessment, in my opinion there is a lack of discussion and/or comparison with previous works on the subject on Vesuvius or elsewhere. For example, it would be interesting to compare with a) traditional (field studies) methodologies for lahar hazards, how the quantitative probabilities given in this work can translate (or not) to the typical high, medium, low quantitative probabilities of hazard maps b) similar previous studies, in particular with Tierz et al, 2017. How it compares in terms of the physical model used, how probabilities are obtained and analysis of final results.

   *We would not dive into categorizing the probabilities into "high", "medium", and "low" as this implies some sort of "decision" that we do not feel comfortable with. However in the Results section we have now compared our results with some of the findings in Tierz et al (2017) (e.g., Lines 252-254; 295; 299-300; 362-366).*

I think these additions can be incorporated easily and without substantially altering the original structure of the draft, but I think they will improve the impact among other people that study lahars.

**EC1**: 'Comment on egusphere-2023-1295', Virginie Pinel, 05 Oct 2023 reply

Two reviewers underlined the scientific interest of the manuscript but raised several points that should be addressed before publication. In particular, I would encourage the authors to make better reference to previous work on understanding lahars and assessing the associated risks, and to enrich the discussion on the potential application of this methodology to other volcanic areas.

In addition, I would also suggest improving the figure captions. In particular, Figure 1 refers to "four illustration points" when only two are shown. It should be explained when information is displayed on a shaded topography (which is not directly a DEM since no direct altitude information is displayed on the image...).
*We thank the Editor, we have corrected accordingly the caption.*

I suggest that the authors respond to all these comments before submitting a revised version of your manuscript, which should then represent a significant contribution to the field.